# An Evaluation of Type 1 Interferon Related Genes in Male and Female-Matched, SARS-CoV-2 Infected Individuals Early in the COVID-19 Pandemic

**DOI:** 10.3390/v16030472

**Published:** 2024-03-20

**Authors:** Tom P. Huecksteadt, Elizabeth J. Myers, Samuel E. Aamodt, Shubhanshi Trivedi, Kristi J. Warren

**Affiliations:** 1Salt Lake City VA Medical Center, Salt Lake City, UT 84148, USA; tom.huecksteadt@hsc.utah.edu (T.P.H.); elizabeth.myers@utah.edu (E.J.M.); s.aamodt@utah.edu (S.E.A.); shubhanshi.trivedi@hsc.utah.edu (S.T.); 2Department of Neurology, University of Utah, Salt Lake City, UT 84132, USA; 3Department of Internal Medicine, Pulmonary Division, University of Utah Health Science Center, Salt Lake City, UT 84132, USA; 4Division of Infectious Diseases, University of Utah, Salt Lake City, UT 84132, USA

**Keywords:** COVID-19, SARS-CoV-2, interferon stimulated genes (ISG), IFN*α*, IFN*γ*, IL-6, toll-like receptor (TLR), Lipopolysaccharide (LPS), innate lymphoid cells (ILC), group 2 ILC (ILC2), group 3 ILC (ILC3), natural killer cells (NK), pattern recognition receptors (PRR), pathogen associated molecular patterns (PAMP)

## Abstract

SARS-CoV-2 infection has claimed just over 1.1 million lives in the US since 2020. Globally, the SARS-CoV-2 respiratory infection spread to 771 million people and caused mortality in 6.9 million individuals to date. Much of the early literature showed that SARS-CoV-2 immunity was defective in the early stages of the pandemic, leading to heightened and, sometimes, chronic inflammatory responses in the lungs. This lung-associated ‘cytokine storm’ or ‘cytokine release syndrome’ led to the need for oxygen supplementation, respiratory distress syndrome, and mechanical ventilation in a relatively high number of people. In this study, we evaluated circulating PBMC from non-hospitalized, male and female, COVID-19+ individuals over the course of infection, from the day of diagnosis (day 0) to one-week post diagnosis (day 7), and finally 4 weeks after diagnosis (day 28). In our early studies, we included hospitalized and critically care patient PBMC; however, most of these individuals were lymphopenic, which limited our assessments of their immune integrity. We chose a panel of 30 interferon-stimulated genes (ISG) to evaluate by PCR and completed flow analysis for immune populations present in those PBMC. Lastly, we assessed immune activation by stimulating PBMC with common TLR ligands. We identified changes in innate cells, primarily the innate lymphoid cells (ILC, NK cells) and adaptive immune cells (CD4+ and CD8+ T cells) over this time course of infection. We found that the TLR-7 agonist, Resiquimod, and the TLR-4 ligand, LPS, induced significantly better IFNα and IFNγ responses in the later phase (day 28) of SARS-CoV-2 infection in those non-hospitalized COVID-19+ individuals as compared to early infection (day 0 and day 7). We concluded that TLR-7 and TLR-4 agonists may be effective adjuvants in COVID-19 vaccines for mounting immunity that is long-lasting against SARS-CoV-2 infection.

## 1. Introduction

SARS-CoV-2 arose as a novel coronavirus in late 2019 and was eventually identified as the causative agent of the systemic and pulmonary syndrome now called COVID-19 [1,2]. As infection rates increased from early 2020–2022, the research and clinical communities came together to try to understand the pulmonary manifestations that led to hospitalization in certain individuals but not others [3,4,5]. Two characteristics were identified regarding immunity to SARS-CoV-2 infection in humans, both of which explain why COVID-19 proved complicated to treat. Early clinical data showed reduced numbers of many lymphoid populations, including innate lymphoid cells and T cells, with increased neutrophils and inflammatory monocytes, suggesting an imbalanced, non-specific immune response in the COVID-19+ population [6,7]. Along this same line of reasoning, human cytokine syndrome, previously called the cytokine storm, was reported in many individuals as they moved into critical illness with COVID-19 [8,9]. Like other coronaviruses, SARS-CoV-2 significantly evades early detection by interrupting anti-viral, type 1 interferon programming in humans [10,11,12]. There are significant data that show that type 1 interferons shape and refine the adaptive immune responses in respiratory viral infections [13,14]. Together, these characteristics, in conjunction with unique comorbidities (obesity, diabetes, pulmonary hypertension), caused many of the complications with COVID-19 that are not well treated in the clinic [1,15,16,17,18,19]. In addition, biological sex was associated with increased severity of COVID-19 in males versus females, and a long-term COVID-19 sequela, now called ‘long COVID’, is more common in females compared to males. All these characteristics suggest an aberrant immune response to this novel respiratory virus [6,20,21].

In the current study, we chose to evaluate innate immunity in PBMC from COVID-19+ individuals with the goal of understanding innate lymphoid cells in anti-viral responses to SARS-CoV-2 infection. Innate lymphoid cell (ILC) subsets resemble the CD4+ T helper cells and CD8+ cytotoxic T cells in the role they play in the immune response [22,23,24]. ILC1 and NK cells are the counterparts of the CD4+ IFNγ+ and CD8+ cytotoxic T cells [22,25], respectively, while ILC2 resemble the CD4+ IL-4/IL-13 producing Th2 cells, with the caveat that ILC2 only produce IL-4 under specific activation with leukotrienes [26,27]. Lastly, ILC3 are RORγT+ innate lymphoid cells, that like Th17 cells, produce IL-17 to program neutrophilic responses in the lung [28,29]. T cells have been well studied for decades in a variety of pulmonary illnesses, but while there are similarities between T cells and ILC, ILC diverge from T cell responses in a few ways. They tend to reside along epithelial barriers in the lung, where they can respond to cytokine and pathogen-associated molecular patterns (PAMPs) via pattern recognition receptor (PRR) activation [30,31]. ILC respond within hours versus days to pulmonary insults in comparison to T cells, and they do so without antigen specificity as they do not have T cell receptors. ILC are fewer in number, but they produce cytokine at a much higher rate in comparison to their T cell counterparts. These cytokines arise to program the initial immune responses based on the type of assault the host is experiencing. Lastly, while hypothetically replenished and recruited from the bone marrow by hematopoiesis during an active infection, it is unclear whether these cells are actively increased in circulation during SARS-CoV-2 infection to replenish the innate lung immune response [32,33].

Our study objective was to investigate anti-viral, interferon-stimulated gene networks (ISGN) in circulating immune cells over the time course of SARS-CoV-2 infection. We focused on the type 1 and type 2 interferon responses in our cohort. The type 1 interferon family, including interferon (IFN) α and IFNβ, are utilized solely for the purpose of reducing viral replication and release, thereby limiting the progression of viral-associated diseases in delicate lung tissue. Like many viruses the coronaviruses have adapted means of circumventing this innate arm of the immune response. Interferon proteins induce approximately 800 genes in the Interferon (IFN) Stimulated Gene Network (ISGN) downstream of nod-like receptor (NLR) and toll-like receptor (TLR) pathways [34]. We have developed a panel of approximately 30 anti-viral genes to make an early assessment of the ISGN in the COVID-19+ patient population. All our studies were stratified by male versus female responses as certain innate immune populations demonstrate increased immune activation in females compared to males [35,36,37,38,39,40]. In these studies, we (1) define the composition of innate cell subsets isolated from COVID-19+ people and healthy controls by flow cytometry, (2) identify specific gene expression profiles using RT-PCR, and finally (3) activate immune cells ex vivo with PRR agonists to assess IFN production.

## 2. Materials and Methods

### 2.1. Recruitment and BSL-2 Laboratory Protocols

Hospitalized patients were identified after admissions into the University of Utah hospital or after arriving on the ICU floor for cases of severe COVID-19 infection. Confirmation testing was completed on all COVID-19-positive individuals for active SARS-CoV-2 infection. All samples were deidentified and given a unique study ID. Healthy PBMC samples were collected from ARUP Blood Services located in the University of Utah Hospital. Healthy samples were tested for SARS-CoV-2 to verify the absence of respiratory viral infection in healthy patient controls. All study protocols were reviewed and approved by the Institutional Review Board at the University of Utah (IRB# Middleton, IRB# Spivak). The Institutional Biosafety Review at the University of Utah reviewed and approved BSL-2 protocols, and the Research Safety Committee at the Salt Lake City VA Medical Center reviewed and approved protocols for these studies.

### 2.2. Blood Collection and Peripheral Blood Mononuclear Cell Isolation

One 10 mL tube of whole blood was collected in an EDTA-treated tube for PBMC isolation. 10 mL of blood was diluted with an equal volume of 1X Dubelcco’s PBS and then layered onto an EasySep50 tube (Stemcell Technologies, Vancouver, BC, Canada) filled with 15–17 mL of LymphoPrep (Stemcell). After layering, these tubes were centrifuged at 2000× *g* for 10 min at room temperature. Mononuclear cells were then poured into a clean 50 mL tube, where they were resuspended in 35 mL of Easy Separation Buffer (Stemcell) and centrifuged at 400× *g* for pelleting. Cell pellets were collected after centrifugation and resuspended in cryopreservation media composed of 55% charcoal-scrubbed FBS, 35% DMEM media, and 10% DMSO (ATCC). Cells were stored at 20 × 10^6^ cells/mL at −150 °C until they could be cultured for immune activation and PCR array analysis.

### 2.3. PBMC Culture and Activation Assays

Aliquoted PBMC were quickly defrosted in a 37 °C water bath and resuspended in 10 mL of complete RPMI culture media (10% FBS, 1% Penicillin/Streptomycin, HEPES, and l-glutamine) followed by centrifugation at 400× *g* at room temperature. Cells were counted and placed in culture at a rate of 1 × 10^6^ cells per well in a 24-well cell culture dish. Total ILC were enriched using a human Pan-ILC enrichment kit (Stemcell). Activating cytokines were added to complete RPMI culture media (10% FBS, 1% Penicillin/Streptomycin, HEPES, and l-glutamine) at 10 ng/mL (IL-23, IL-33, IL-7, SCF) or 50 ng/mL (IL-2, IL-3). All cytokines were purchased from Preprotech. For TLR activation assays, Poly I:C (25 μg/mL), LPS (1 μg/mL), or Resiquimod (5 μg/mL) were added to complete RPMI for a total of 48 h to assess immune activation. Cell culture supernatants were collected and stored at −80 °C until immune inflammatory proteins could be measured.

### 2.4. Flow Cytometry

An aliquot of PBMC from each donor was quickly defrosted in a 37 °C water bath and resuspended in FACS stain buffer (BD Pharmingen, San Diego, CA, USA). After resuspension, cells were centrifuged at 400× *g* for 10 min to pellet immune cells for FACS analysis. All pellets were treated with anti-CD16/32 (Cat#564219, BD Biosciences, Franklin Lakes, NJ, USA) for 10 min, then stained with anti-human antibodies (Table 1). Cells were fixed, and data were acquired on the Cytek Aurora within 4 h of fixation. Single-cell color controls were prepared using UltraComp beads (ebioscience), unstained cells, and Zombie aqua viability dye (Invitrogen-Thermo Fisher, Carlsbad, CA, USA). Only stained cells were included to properly compensate and gate the various immune populations detected.

### 2.5. Quantitative RT-PCR

Up to 10^7^ PBMC were aliquoted into RNA lysis buffer, where RNA could be purified for analysis by RT-PCR. As before, RNA was cleaned up with the Zymo Research Mini RNA Isolation kit (Genessee Science, El Cajon, CA, USA). Next, 500 ng of purified RNA was placed into a cDNA reaction (qPCRBIO cDNA Synthesis Kit, Cat # 17-700B) and run under the following cycling conditions: 42 °C for 30 min, 85 °C for 10 min, hold at 4 °C on an ABI GeneAMP PCR System 9700 Thermocycler. Primers were commercially available from SABiosciences and used according to the manufacturer’s instructions with PowerUp SYBRGreen Master Mix (Cat# A25742, Applied Biosystems, Carlsbad, CA, USA) and run using preset cycling conditions for SYBRGreen on the QuantStudio3 PCR instrument. Changes in gene expression were determined using the 2^ΔΔCt^ method.

### 2.6. Protein Assays and Cytokine Detection

According to the manufacturer’s instructions, inflammatory proteins were detected in cell culture supernatants using the Quansys Biosciences Q-Plex Human Cytokine Release Syndrome (16-plex) protein array (Logan, UT, USA). Cell culture supernatant was precleared by centrifugation at 10,000× *g*, then diluted 1:2 with assay diluent as directed in the kit instructions. Quantitation of protein per milliliter of cell culture fluid was determined using a 4-parameter standard curve. Results for these assays are displayed as Mean ± SEM.

## 3. Results

### 3.1. Immune Cells from COVID-19+ Individuals Make Appreciably More Inflammatory Cytokines on Day 28 of SARS-CoV-2 as Compared to Day 0 or Day 7

Male and female individuals were enrolled on the day they were diagnosed with COVID-19 (day 0) for blood draws at three time points: day 0, 7, and 28 post-diagnoses. Healthy, non-infected individuals were included as controls, and individuals hospitalized with COVID-19 were recruited during their hospital stay for a single blood draw. Comparatively, we evaluated the total number of CD45+ cells and neutrophils between groups. We found a significant increase (d0 vs. d28; *p* < 0.05) in viable CD45+ leukocytes by day 28 after diagnosis (Figure 1A) and an increase in circulating neutrophils at day 0 of infection compared to healthy controls (Figure 1B; # *p* < 0.05). Next, to assess immune activation, 1 × 10^6^ CD45+ cells from each group were stimulated with PMA/Ionomycin for 24 h, followed by measurements of IL-1β, TNFα, and IL-6 (Figure 1C–E). Healthy control leukocytes made negligible levels of IL-1β, TNFα, and IL-6, whereas CD45+ cells from hospitalized individuals made a significant amount of IL-6 but not IL-1β or TNFα. In general, we noted the highest levels of cytokines from day 28 infected individuals as compared to day 0 or day 7. At day 28, CD45+ cells produced significantly higher amounts of IL-1β, TNFα, and IL-6 as compared to day 0 or day 7. Although we initially hypothesized that biological sex would be an important factor in the level of immune cells detected in circulation or inflammatory protein production, we found no effect of biological sex on these experimental outcomes.

### 3.2. Characterization of Innate Lymphoid Cells in the Blood of COVID-19+ Individuals to 4 Weeks of SARS-CoV-2 Infection

PBMC were isolated from de-identified patients on the day of diagnosis with SARS-CoV-2 infection (day 0), 1 week later (day 7), and 4 weeks after diagnosis (day 28). In these experiments we assessed innate (Figure 2) and adaptive lymphoid immune populations (Appendix A) by flow cytometry, as reported previously [41]. We were able to identify Total ILC as lineage-negative (Lin-) CD127+ cells and group 2 innate lymphoid cells (ILC2) as CD45+ Lin- CD25 and CRTH2+ cells in total PBMC. Group 3 innate lymphoid cells (ILC3) were identified as CD45+ Lin- CD25+NKp46+ and ICOS+, and NK cells were identified as CD25+NKp46+ and ICOS- [42,43]. Group 1 innate lymphoid cells were not identified as it is now established that this population of innate lymphoid cells in total is tissue localized. Innate lymphoid cells resemble the CD4+ T helper cell and CD8+ cytotoxic T cell subsets that have been well-studied for decades in a variety of pulmonary illnesses [22,44]. Over the course of SARS-CoV-2 infection, we hypothesized that there would be steady increases from day 0 to day 7 in innate lymphoid cell populations in the COVID-19+ individuals compared to healthy controls. We show here that while Total ILC (Figure 2A) were not different between groups, ILC2 from COVID-19+ individuals were lower on day 0 compared to the levels of ILC2 detected in healthy controls (Figure 2B). ILC3 were increased at day 0 (Figure 2C) compared to healthy controls. The percentage of CD45+ cells, identified as natural killer cells, decreased at day 0, day 7, and day 28 of infection compared to healthy controls (Figure 2D). NK cells made up the largest portion of ILC. Importantly, no significant differences in any ILC population were determined between the biological sex although we did initially anticipate changes in ILC2 given that these cells have been shown to vary by sex in allergic asthma [45]. We furthermore enriched total ILC from SARS-CoV-2 infected individuals at each time point and found that if we activated these cells with IL-2, IL-7, IL-33, and IL-23, we detected an increased amount of IFNγ production by innate lymphoid cells isolated from day 28 infected individuals. Interestingly, IFNγ was highly produced by day 28 ILC without cytokine stimulation as compared to day 0 with cytokine stimulation (Figure 2E). Additional ILC specific cytokines were examined, including IL-5, IL-13, and IL-17, and these cytokines were not significantly induced in culture by IL-23 and IL-33.

### 3.3. Characterization of Interferons and Interferon-Stimulated Genes in COVID-19+ Individuals

Total PBMC were isolated from COVID-19+ individuals as before with healthy controls included to determine baseline gene expression. A panel of 36 immune-inflammatory and interferon-stimulated genes was evaluated by RT-PCR. Transcripts for IFNG and IFNA1 were increased in COVID-19+ individuals at day 0 and day 7 time points (Figure 3A,C). IFNA2 was elevated in a few COVID-19-positive individuals, but collectively, this did not reach statistical significance (Figure 3B). Gene expression profiling was broken down by day of COVID-19 infection for each individual, and we separated the time-point data by biological sex (Figure 3D). We found that RELA, NFKB2, and TRAF6 reached peak expression by day 7 after diagnosis and later tapered down in most of the samples by day 28. From day 0 to day 7, a significant induction of the inflammasome pathway genes, which included NLRC4, NLRP1, CASP1, PYCARD, and IL-18, was detected in COVID-19+ individuals compared to healthy controls. As there is cross-talk between the inflammasome and interferon-stimulated network in viral infections, we were not surprised to also see IFIT1, IFIH, and DDX58 anti-viral genes upregulated in COVID-19+ individuals. The highest expressed genes by SARS-CoV-2 infection included MAVS, IRF5, and IRF7, and the TLRs (TLR-3, -7, and -9). These highly expressed genes led us to test the responsiveness of immune cells to TLR-agonists in the next experiments to confirm that immune populations are sensitive to the activation of viral mimics.

### 3.4. Type I and Type II Interferons Are Expressed in Late COVID-19 Infection following Stimulation with TLR Ligands

We chose to assess the activation of interferons following stimulation with TLR-agonists as before in COVID-19+ males and females from the day of diagnosis (day 0) to day 28 after SARS-CoV-2 detection (Figure 4). The bacterial ligand, LPS, activates the inflammatory response through the pattern recognition receptor (PRR), TLR-4. We found that when we stimulated PBMC from COVID-19+ donor samples collected on day 28, there was a significant increase in IFNα over PBS stimulation. This pattern of increased IFNα was more pronounced in the females compared to males. We anticipated that LPS would stimulate significant IFNγ production; this reached significance in day 7 stimulated samples for males but not females. However, when day 28 samples were stimulated ex vivo with LPS, we saw a significant increase in IFNγ in both male and female PBMC. Resiquimod induced significant IFNα and IFNγ at day 7 and day 28 compared to PBS in COVID-19+ PBMC. This was significant in both male and female cells. Taken together, this is a strong indication that the immune response is sufficiently activated against future viral and bacterial responses in these non-hospitalized individuals.

## 4. Discussion

SARS-CoV-2 is now a stable viral infection in humans with a broad range of clinical manifestations [2,46]. The ever-changing landscape of SARS-CoV-2 in human hosts requires continuous research to understand how to prevent and treat COVID-19. It is clear that consistent monitoring to determine strain specificity from season to season is necessary and studies that characterize the specific immune responses to the viral variants will direct better vaccine strategies [42,47]. These intelligently designed mRNA vaccines, with masking and good hand hygiene, are the best way to prevent the levels of hospitalizations seen in 2021 [48]. Ultimately, we learned that SARS-CoV-2 recombines rapidly as approximately four variants, with subvariants, have arisen since the beginning of the pandemic [46]. With the identification of each variant, we learned that certain variants caused greater disease severity while other variants spread more quickly but did not lead to as many critical care cases as before [49]. Early data from the pandemic showed that male COVID-19-positive individuals progress to severe SARS-CoV-2 infection more often than females [50,51,52,53]. This project was partially prompted by this observation. It is commonly understood that female hosts mount better innate immune responses to infection in comparison to males. In this way, most respiratory infections are cleared more quickly in females; however, these increased anti-viral inflammatory responses occur at the expense of delicate lung tissue, which limits vital gas exchange and lung function. This was not the case with male versus female responses to SARS-CoV-2, as initially, severe COVID-19 was less likely in females, and male COVID-19+ individuals had higher rates of mortality in comparison to females [52,53,54,55]. Acute respiratory distress syndrome (ARDS) and sepsis were common in high-risk COVID-19+ individuals (predominantly male), highlighting a need to reprogram immunity towards a more productive anti-viral response [55,56] that prevents excess inflammation and improves adaptive immunity and T cell memory [13,14,57]. Disappointingly, when we evaluated biological sex variables in our analysis, we did not find clear statistical significance in our experimental outcomes. This may be related to sample size, and not knowing the exact date of initial SARS-CoV-2 infection in our cohort. This may explain why we only saw a subset of females with IFNA and IFNG transcripts trending upwards compared to males (Figure 4).

Because we did not have the date of symptom onset for hospitalized patients, and we were greatly limited by the reduced cellularity in blood samples collected from the hospitalized COVID-19+ individuals, we did not fully assess immune activation. Only one experimental output was achieved and showed increased IL-6 production, but not IL-1β or TNFα, by PBMC from hospitalized individuals, with confirmation of immune cytopenia in circulation (Figure 1) [58,59]. It is likely that without medical intervention in the form of corticosteroids to reduce inflammatory responses to SARS-CoV-2 infection, those severe COVID-19+ patients’ immunopathologies would remain unchecked [60]. Lastly and probably most importantly, circulating immune populations in the blood are only a window into the lung inflammatory response [61]. This certainly is a limitation of the current study. It is intriguing to propose future studies with experimental animal models that could evaluate innate lymphoid populations and tissue-resident macrophages in the lung during an active SARS-CoV-2 infection [62,63]. Hypothetically, early responses in lung ILC and macrophages are likely modified by this coronavirus to evade detection and prolong infection in hosts.

As briefly touched upon before, the innate immune response is important for long-term adaptive SARS-CoV-2-specific immunity. We undertook this study to see if we could identify any unique characteristic immune responses in this patient population, with our primary focus on the interferons and the ILC populations. We consistently found a steady increase in inflammatory responses to activating cytokines in total ILC (Figure 3) and in our total CD45+ cells at day 28 after SARS-CoV-2 detection (Figure 1). Whether immune cells were non-specifically activated, activated with cytokines, or activated with a TLR-agonist, we observed the greatest number of cytokines later in the resolution phase of the infection. Resiquimod induced the highest cytokine production in ex vivo stimulated PBMC from the COVID-19+ outpatient population at day 28. This was a critical piece of data as immune-boosting strategies against SARS-CoV-2 infection may benefit from including this TLR-7/8 agonist [64,65]. Whether this works in conjunction with the current mRNA COVID-19 vaccines is not known, but interesting to postulate. In summary, there is still a need for research directed at improving clinical outcomes. In late 2023, we are still seeing new and now repeat cases of COVID-19 in our patient populations, with much of the latest literature concluding that we still need to do more to prevent or prepare for future pandemics.

## Figures and Tables

**Figure 1 viruses-16-00472-f001:**
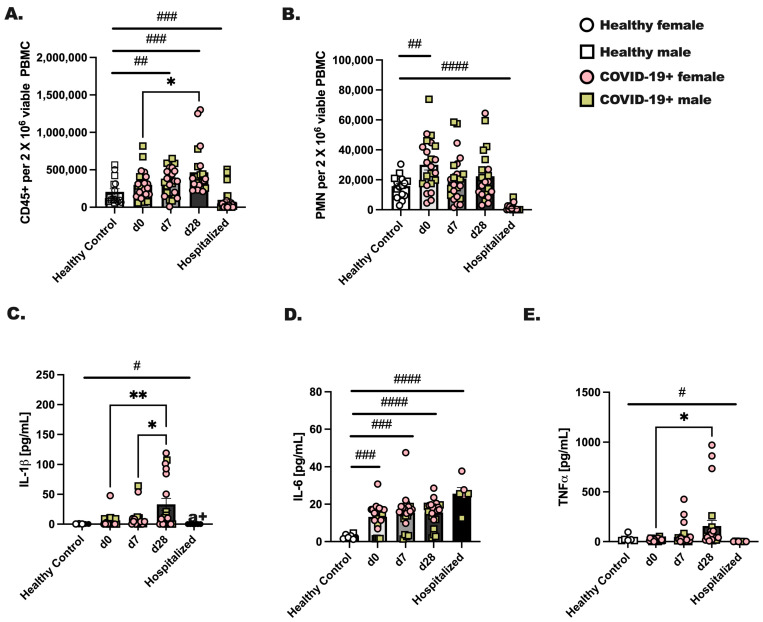
Immune cells from COVID-19+ individuals make appreciably more inflammatory cytokines at day 28 of SARS-CoV-2 as compared to day 0 or day 7. Whole blood was collected from healthy controls and SARS-CoV-2 infected (COVID-19+) people at three time points. Day 0 represents a blood draw on the day of SARS-CoV-2 detection, day 7 is 1 week later, and day 28 is 4 weeks after SARS-CoV-2 detection. Hospitalized patients were all positive for SARS-CoV-2, and blood was collected during their hospital stay. (**A**) CD45+ viable cells and (**B**) neutrophils were detected by flow cytometry for each group and time point. (**C**–**E**) CD45+ cells were stimulated with PMA (25 ng/mL) and Ionomycin (1 μg/mL) to stimulate cytokines, which were measured after an overnight stimulation. (**C**) IL-1β, (**D**) IL-6, and (**E**) TNFα were detected in culture supernatants by protein array. Statistical analysis was completed using a two-way ANOVA and Tukey’s multiple comparisons. * Indicates a *p* = 0.05 or less and ** less that *p* = 0.01. # indicates a significant difference (*p* < 0.05) by Student’s *t*-test between indicated bars; # *p* < 0.05, ## *p* < 0.01, ### *p* < 0.001, #### *p* < 0.0001. Healthy controls—n = 18, and for COVID-19+ individuals, n = 22. Throughout the manuscript, an open square represents a male, non-infected donor, an open circle represents a female, non-infected donor, a colored (pink) circle represents a COVID-19+ female participant, and a colored (yellow/tan) square represents a COVID-19+ male participant.

**Figure 2 viruses-16-00472-f002:**
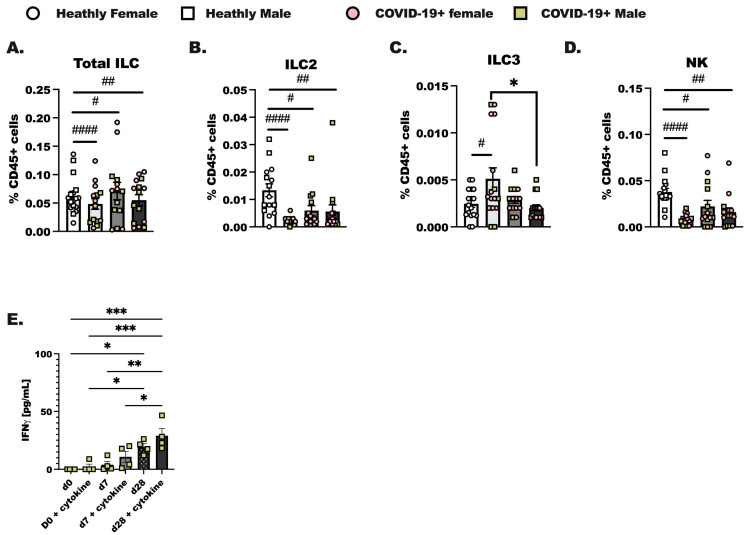
Characterization of innate lymphoid cells in the blood of COVID-19+ individuals out to 4 weeks of SARS-CoV-2 infection. Healthy controls and COVID-19+ samples were collected as before (Figure 1); hospitalized patient samples were not included as most of these individuals experience lymphopenia, and ILC were not readily detected by flow. (**A**) Total innate lymphoid cells (ILC) were identified as lineage negative (Lin- (CD3-, CD19-, CD11c-, CD11b-, CD15-, TCR *αβ*-, TCR *γδ*-, Ter-119, and Ly-76)) following by inclusive gating for CD45 and CD127. (**B**) Group 2 innate lymphoid cells (ILC2) were CD45+ Lin- and CD127+, with additional gating on CD25+ and CRTH2+ cells. (**C**) Group 3 innate lymphoid cells (ILC3) were gated as CD45+ Lin- CD127+, followed by gating for CD25, NKp46+, and ICOS-positive cells, whereas (**D**) natural killer cells (NK) were CD25+ NKp46+, and ICOS-. (**A**–**D**) represents each population as a percentage of the viable CD45+ cells. (**E**) Interferon (IFN) γ production by enriched ILC cultured with the ILC prosurvival cytokines, IL-2 and IL-7, and the ILC activating cytokines (IL-23, IL-33) was assessed using a protein array. Statistical differences were determined in the same way as before using a two-way ANOVA with post-testing to make between COVID-19+ groups (d0 vs. d7, d7 vs. d28, d0 vs. d28 by post-test) comparisons. As before, * indicates a *p* = 0.05 or less, ** less than *p* = 0.01, and *** indicates *p* < 0.001. A Student’s *t*-test was used to compare the healthy control group to each COVID-19+ group; # *p* < 0.05, ## *p* < 0.01, #### *p* < 0.0001 by *t*-test. For healthy controls, n = 15, and for COVID-19+ individuals, n = 14.

**Figure 3 viruses-16-00472-f003:**
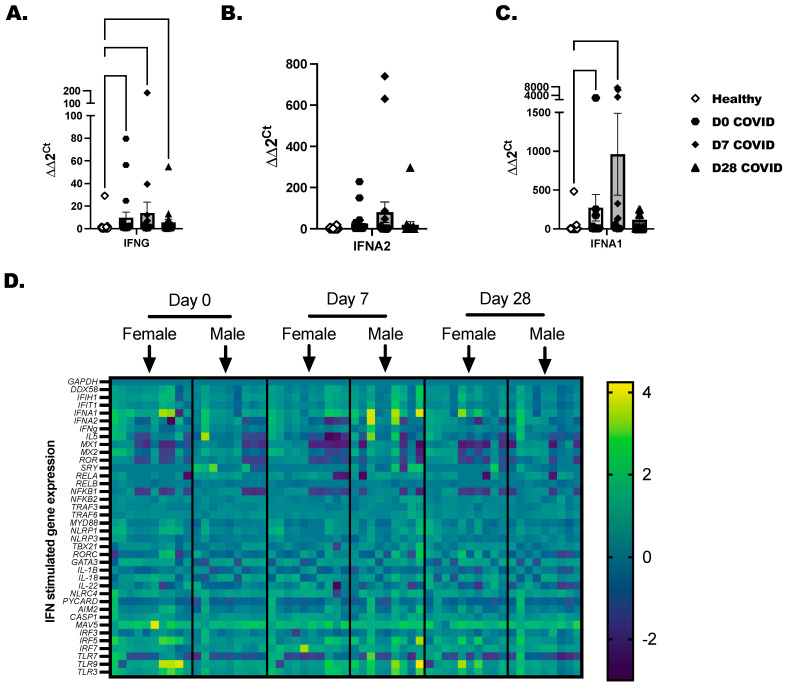
Characterization of interferons and interferon-stimulated genes in COVID-19+ individuals. Healthy and COVID-19+ donor-derived PBMC were lysed, and RNA was collected and purified. Purified RNA was tested by RT-PCR for a panel of 32-interferon-related genes. (**A**) IFNG, (**B**) IFNA2, and (**C**) IFNA1 are independently shown as bar graphs with the mean ± the SEM of fold-change detected for each COVID-19 individual compared to the sex-matched control. Male COVID-19+ individuals were compared to male non-infected, and Female COVID-19+ were compared to female non-infected as baseline controls. The baseline controls were randomly selected and included with every PCR plate. (**D**) Representative heatmaps that show mean log fold-change across groups. Scale represents genes modified in comparison to over controls. Statistical significance and indicators are as described before (Figure 1).

**Figure 4 viruses-16-00472-f004:**
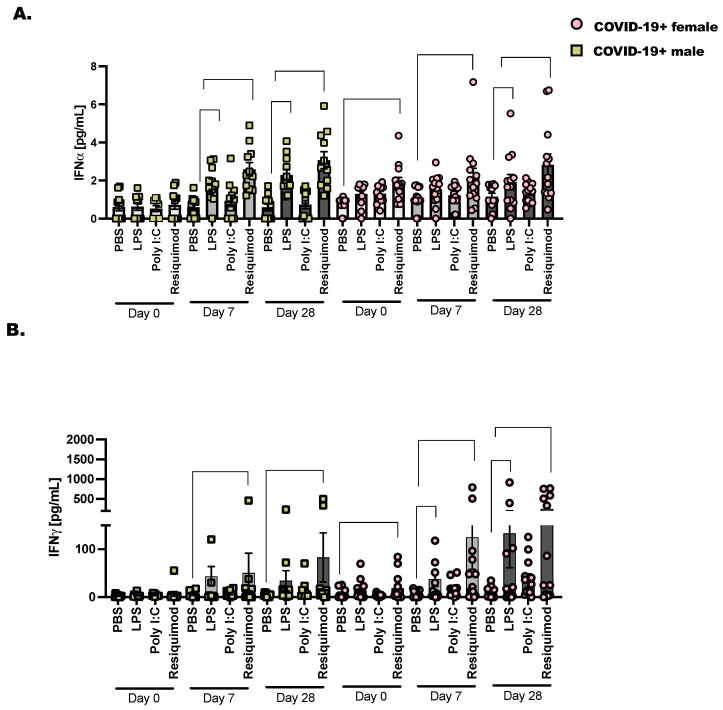
Type I and type II interferons are expressed in late COVID-19 infection following stimulation with TLR ligands. As detailed before (Figure 1), 2 × 10^6^ CD45+ cells were cultured with lymphoid and myeloid, pro-survival cytokines (IL-2, IL-7, and IL-3—10 ng/mL each) followed by stimulation for 48 h with PBS as control, LPS (1 μg/mL), Poly I:C (25 μg/mL) and Resiquimod (5 μg/mL). Cell culture supernatants were collected and pre-cleared at 10,000× *g* before diluting (1:2) and adding them to the 96-well protein array plate. (**A**) Interferon *α* and (**B**) Interferon *γ* are displayed as mean pg/mL ± the SEM for each donor and timepoint. Healthy controls were selected at random, and at least one healthy male or female baseline control was included per plate analyzed. Statistical significance and indicators are described in Figure 1. Two-way ANOVA and Tukey’s multiple comparisons.

**Table 1 viruses-16-00472-t001:** Flow cytometry antibody list.

Antigen	Color	Clone	Source
CD3ϵ	Alexa Fluor 647	SK7	Biolegend: Cat# 344825
CD19	BUV615	SJ25C1	BD Biosciences: Cat# 612990
CD123	BV785	6H6	Biolegend: Cat# 306032
CD11c	Alexa Fluor 700	B-ly6	BD Biosciences: Cat# 561352
CD11b	BV750	D12	BD Biosciences: Cat# 747210
CD4	BUV737	L200	BD Biosciences: Cat# 749213
CD8	BUV395	RPA-T8	BD Biosciences: Cat# 563796
TCR γ/δ	FITC	eBioGL3	eBiosciences: Cat# 11-5711-82
CD278	BV421	DX29	BD Biosciences: Cat# 562901
HLA-DR	APC-Fire 750	L243	Biolegend: Cat#307658
CD127	PE-eFluor 610	eBioRDR5	eBiosciences Cat# 61127842
CD25	PE	BC96	Biolegend: Cat# 302606
Ter-119	APC	TER-119	eBiosciences Cat# 17-5921-82
CD15	BV650	HI98	BD Biosciences: Cat# 564232
CD16	BUV496	3G8	BD Biosciences: Cat# 612944
CD117	BV605	104D2	Biolegend: Cat# 313218
CD45	BUV805	HI30	BD Biosciences: Cat# 612892
CD161	eFluor 450	HP-3G10	eBiosciences Cat# 48-1619-42
CRTH2	PerCP-Cy5.5	BM16	Biolegend: Cat#350116
NKp46	PE-Cy7	9E2/NKp46	BD Biosciences: Cat# 562101

All antibodies were used according to recommended dilutions from the manufacturer for a single test (single test = up to 5 × 10^6^ cells).

## Data Availability

All data are contained within the article.

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
