# Peer review of "An Evaluation of Type 1 Interferon Related Genes in Male and Female-Matched, SARS-CoV-2 Infected Individuals Early in the COVID-19 Pandemic"

_viruses, 2024, doi:10.3390/v16030472_

Round 1
Reviewer 1 Report
Comments and Suggestions for Authors
The manuscript by Huecksteadt and colleagues evaluates the circulating subsets of innate lymphoid cell (ILC) subsets and the expression of a subset of type I interferon genes in 22 male and female SARS-CoV-2-infected individuals at the beginning of the COVID-19 pandemic. In the experimental design, individuals who tested positive for the virus were evaluated at three time points: day 0 (when they were diagnosed); day 7, and day 28 post-diagnosis. The major conclusion from this manuscript is by 4 weeks post-infection IFN-γ levels are significantly higher after infection. While the conclusions of the authors fit the data presented, several issues need to be addressed.
Comments:
1. This study examines a group of female and male individuals who tested positive for the SARS-CoV-2 virus. Many details are missing such as a) the range of ages of individuals; b) the length of their illness; c) and how many SARS-CoV-2 infected individuals eventually developed a serious illness. Such data is important as the infection in younger individuals may not be as serious as in older individuals with co-morbidities.
2. Line 123: “1e6 cells” should be changed to “1 x 106 cells.”
3. Table 1. This table lists the antibodies to the different markers used in the study. It would be helpful to the reader if the cells that express these markers were also listed in the table.
4. Figure 1, panels A and B: The number of viable cells in these two panels should be stated as per X number of PBMCs.
5. line 192: the following, “(C) TNF𝛼, (D) IL-1𝛽 and (E) IL-6” should actually be “(C) IL-1𝛽 and (D) IL-6 and (E) TNF𝛼.” Also on panel C, y-axis: Should “IL-1b” be changed to “IL-1β?”
6. Lines 215-217: The following statement should be reworded. “We show here that while Total ILC (Figure 215 2A) were not different between groups, ILC2 were reduced in circulation on day 0 (Figure 216 2B and 2F).” The problem here is the authors state that “ILC2 were reduced in circulation on day 0.” As the authors did not analyze samples before infection, so they can’t make that statement. However, they can state that the levels of ILC2 cells were less than in the healthy controls. Please reword.
7. Figure 2: In this figure, the authors have headings for panels A to D. To be consistent, they should also have headings for panels E-H.
8. Figure 3, lines 273-274: The statement “(A) IFNG, (B) IFNA2, and (C) IFNA1 are shown independently bar graphs with the mean ± the SEM of fold-change detected for each COVID-19 individual compared to sex-matched control” should probably be changed to “(A) IFNG, (B) IFNA2, and (C) IFNA1 are independently shown as bar graphs with the mean ± the SEM of fold-change detected for each COVID-19 individual compared to sex-matched control.”
9. Figure 4. In this experiment, the authors examine IFN-alpha and IFN-gamma following treatment with pro-survival cytokines. The authors analyze the data for statistical significance. First, the data is difficult to interpret with the number of samples. Secondly, the authors group the males and females. The male and female data should be on separate graphs since the authors state in lines 89-91 that, “All our studies were stratified by male versus female responses as certain innate immune populations demonstrate increased immune activation in females compared to males.”
10. Another concern I have with Figure 4 is whether the levels of IFN-α are biologically significant. As an example, the levels of IFN-α at day 28 go from ~1 pg/ mL (PBS control) to 2 pg/mL (day 28).
11. lines 358-359: The following sentence, “Resiquimod induced the best immune responses in the COVID outpatient population at day 28,” could be interpreted as the outpatients were being treated with Resiquimod. Perhaps the authors should try revising to, “Resiquimod induced the best immune responses from cell populations ex vivo from the COVID outpatient population at day 28.”
Comments on the Quality of English Language
The English is fine.
Author Response
Please see attachment for our response to reviewer 1.

Reviewer 2 Report
Comments and Suggestions for Authors
Huecksteadt et al have studied immune response of PBMCs from COVID infected individuals with focus on interferon and interferon stimulated gene signature. The study is overall well designed with the goals well defined and results well interpreted. However, the manuscript needs formatting in several aspects. Interestingly there is no author with affiliation - 4. My comments are as follows:
1. The focus on ILCs in the early part of the paper is not justified. What was the reason for looking at ILCs. Also the conclusions of the ILC results are not mentioned in the abstract and have not been discussed for what the possible ramifications might be.
2. References for the entire ILC section (lines 58-78) are missing.
3. Line 93 - RNA microarray has been mentioned although the authors did RT-PCR.
4. Line 175 - Why was PMA/Ionomycin used as a stimulus and IFN levels not assessed? The authors should clarify. The d28 cells respond better to PMA/ionomycin too. What are the implications of this observation? Are the cells better responders irrespective of the stimulus?
5. Line 212 - Reference required.
6. The authors should comment on why IFN-beta was not evaluated in the study.
7. Figure 4 - The authors should look at other inflammatory cytokines as studied in figure 1 to comment on the specificity of the response. Hence in the abstract the authors should state that the cells at d28 produce higher IFNa and IFN gamma instead of broadly stating it as IFN response.
Do healthy controls show a response in terms of IFN?
8. What is the source of IFN-gamma in response to TLR stimulation in figure 4. Is it assumed that T cells are the primary source even in absence of concurrent TCR stimulation as shown previously (PMID: 26205220)? This aspect needs to be discussed. Looking at additional cytokines and chemokines will be helpful. Lack of significant response to LPS is justified in that case since human T cells are hyporesponsive to TLR4 stimulation.
9. Have the authors analysed the cellularity in the patients? They only report for number of neutrophils. Differences in other cell populations such as monocytes and T cells should be provided as a table.
Comments about statistical analysis:
1. Figure 1 and related analysis: To more effectively demonstrate that the response increases over time in cells derived from COVID-19 patients, a one-way repeated measures ANOVA should be conducted, focusing solely on the time factor. This statistical approach will clearly isolate the effect of time on immune response without the confounding influences of comparisons to hospitalized patients or uninfected controls. These latter comparisons, while interesting, diverge from the primary objective of assessing time-related changes and could be more appropriately included in the supplementary materials to maintain focus in the main analysis.
2. Figure 2 and related analysis: For panels F, G, and H, where absolute counts of CD45-positive cells are plotted, these should be reconsidered. Variations in the number of cells harvested within each patient need to be accounted for, making absolute counts less relevant. For panels B, C, and D, if the goal is to show time-wise trends in cell type proportions, comparisons to healthy patients are extraneous and should be omitted, as per the suggestion for Figure 1. A more suitable approach might be employing a negative-binomial or an appropriate Generalized Linear Model (GLM) with patient ID as a factor. For panel E, considering some values are close to zero, a log-normal regression could be more appropriate than a two-way ANOVA. Regardless of the chosen method, it's crucial to include patient ID as a factor and to formally test for interaction between 'day number' and 'cytokine stimulation' rather than making multiple individual comparisons between day and cytokine stimulation levels. This approach will provide a more nuanced understanding of the interaction effects.
3. Figure 3 and related analysis: The conversion of delta Ct values to fold change is problematic for several reasons. Firstly, choosing a sample from among the healthy controls for such calculations introduces ambiguity. Secondly, converting to fold change leads to asymmetrical error bars, misrepresenting variability. Thirdly, this conversion necessitates split scales in visual representations, restricting the ability to consolidate all comparisons in one plot.
A preferable approach would be using hierarchical clustering based on the original delta Ct values for creating clustered heatmaps. This method would allow for a clearer visualization of expression patterns across gene families and time points without the distortions introduced by fold change conversion.
4. Figure 4 and related analysis: It would be beneficial to draw a separate line plot for each stimulatory agent (LPS, Poly I:C, and Resiquimod) after normalizing all with respect to PBS. Place days on the x-axis and concentration on the y-axis, focusing exclusively on COVID-19-infected patients. Shift comparisons with healthy controls to the supplementary section to streamline the main analysis (retain only the first time point when doing this). When conducting the one-way ANOVA for generating the main plots, remember to include patient ID as a factor. This approach will allow for a clearer, more focused examination of the stimulatory effects over time within the infected patient group.
Author Response
Please see the attached file with our responses to reviewer 2.
